# The Effect of Menopausal Status, Insulin Resistance and Body Mass Index on the Prevalence of Non-Alcoholic Fatty Liver Disease

**DOI:** 10.3390/healthcare12111081

**Published:** 2024-05-24

**Authors:** Anastasia Ntikoudi, Alketa Spyrou, Eleni Evangelou, Eleni Dokoutsidou, George Mastorakos

**Affiliations:** 1Department of Nursing, University of West Attica, 12243 Athens, Greece; aspyrou@uniwa.gr (A.S.); elevagel@uniwa.gr (E.E.); edokout@uniwa.gr (E.D.); 2Unit of Endocrinology, Diabetes Mellitus and Metabolism, Aretaieion University Hospital, Medical School of Athens, Ethnikon and Kapodistriakon University of Athens, 11528 Athens, Greece; mastorakg@gmail.com

**Keywords:** menopause, non-alcoholic fatty liver disease, insulin resistance, body mass index

## Abstract

Non-alcoholic fatty liver disease (NAFLD) is common and presents in a large proportion—up to 30%—of the global adult female population. Several factors have been linked with NAFLD in women, such as age, obesity, and metabolic syndrome. To extract appropriate details about the topic, we conducted an extensive search using various medical subject headings and entry terms including ‘Menopause’, ‘Non-alcoholic fatty liver disease’, ‘Insulin resistance’, and ‘BMI’. This exhaustive search resulted in a total of 180 studies, among which only 19 were able to meet the inclusion criteria. While most of these studies indicated a significant rise in NAFLD prevalence among postmenopausal women, two did not find strong evidence linking menopause with NAFLD. Moreover, it was observed that women with NAFLD had higher insulin resistance levels and BMIs compared to those without the condition. In summary, it is important to consider specific factors like risk profile, hormonal status, and age along with metabolic components when treating women presenting with NAFLD. There is need for data-driven research on how gender affects the sensitivity of biomarkers towards NAFLD as well as the development of sex-specific prediction models—this would help personalize management approaches for women, who stand to benefit greatly from such tailored interventions.

## 1. Introduction

Up to 30% of adults worldwide are affected by non-alcoholic fatty liver disease (NAFLD), making it the primary cause of liver disease. NAFLD can be categorized into two main subtypes: non-alcoholic fatty liver (NAFL), also known as simple steatosis, which is a non-progressive form of the disease that rarely advances to cirrhosis, and NASH, the most prevalent progressive form of NAFLD, which leads to cirrhosis and hepatocellular liver disease [1,2]. NASH is characterized by the presence of steatosis, ballooning, and lobular inflammation in liver histology, with or without perisinusoidal fibrosis.

The occurrence and advancement of NAFLD/NASH in women have been linked to fluctuations in reproductive hormones. Women who have polycystic ovary syndrome (PCOS) or experience estrogen deficiency face a heightened susceptibility to NAFLD/NASH. Additionally, older women exhibit higher mortality rates in comparison to men of similar age.

The occurrence of NAFLD varies based on factors like age, gender, and ethnicity. In the general population, the prevalence hovers around 15–25% [3,4]. Women are affected by several factors that contribute to NAFLD, including age, obesity, and metabolic syndrome (MetS) [5]. As individuals age, the likelihood of developing liver disease increases, with higher prevalence observed in women aged 40 to 49, particularly after they have reached the menopause. This suggests that the metabolism of sex steroid hormones may play a role in the development of NAFLD in women [6]. The increased incidence of NAFLD in postmenopausal women can likely be attributed to physiological changes resulting from decreased estrogen levels and alterations in body composition [6,7]. 

The depletion of oocytes leads to menopause, which is characterized by the permanent cessation of menstruation. This, in turn, causes a sharp decline in endogenous estradiol (E2) levels. During the transition to menopause, women typically experience an increase in weight [8]. While age may also play a role in this weight gain, the transitional phase itself is independently linked to a rise in fat mass, particularly in the abdominal area [9]. Additionally, menopausal women tend to experience a decrease in muscle mass and a significant reduction in energy expenditure, primarily due to decreased fat oxidation. These conditions create an environment that promotes an accumulation of total body and visceral fat, without significant changes in energy intake [10].

The accumulation of visceral fat has been found to have detrimental effects on the body, including the production of pro-inflammatory cytokines and free fatty acids, as well as the promotion of oxygen free radicals. These factors contribute to the development of insulin resistance (IR) [11], which plays a crucial role in the pathogenesis of NAFLD. When the body is unable to effectively utilize glucose for energy, compensatory hyperinsulinemia occurs, leading to the utilization of fat as an energy source. This increased lipase activity results in a higher flux of free fatty acids to the liver, ultimately causing steatosis through elevated de novo lipogenesis and reduced lipid export [12].

Research has shown that certain factors, such as body mass index (BMI), age, and menopause, are associated with an increased risk of metabolic syndrome. In postmenopausal women, the interaction between hormonal changes during menopause, the aging process, and central obesity further elevates the risk of developing NAFLD. However, it remains unclear whether insulin resistance in postmenopausal women is directly linked to estrogen deficiency [13,14] or if it is a consequence of aging or other metabolic factors.

Conversely, the menopause is marked by elevated levels of androgens. During this transition, the postmenopausal ovary continues to produce androgens with greater accessibility because of the decline in sex hormone-binding globulin (SHBG). This hormonal shift exacerbates insulin resistance [15].

Steroid hormones play essential roles in regulating a wide variety of physiological functions, including growth, development, reproduction, and metabolism. These chemical messengers are derived from cholesterol and produced through a series of enzymatic reactions. Different tissues in the body can alter steroids at both structural and functional levels—examples include the liver and brain. Steroid hormones can be broadly classified into four main categories: sex hormones, glucocorticoids, mineralocorticoids, and vitamin D. These hormones exert their effects at a cellular level by binding to specific receptors, many of which are nuclear receptors that form dimer complexes upon activation by hormone–receptor binding. The dimer complexes then act as transcription factors for target genes that interact with coregulators—proteins which modulate gene expression either positively or negatively [16].

The delicate equilibrium of lipid levels in the body, known as lipid homeostasis, is regulated through four primary pathways. These pathways include the absorption of lipids from the bloodstream, the synthesis of new lipids, the breakdown of fatty acids, and the export of lipids through very low-density lipoproteins (VLDL) [17]. When there is an imbalance between the acquisition and disposal of lipids, it can result in hepatic steatosis, which is the accumulation of fat in the liver. Research indicates that steroid hormones, particularly estrogen and estrogen receptors (ERs), play a role in the development of NAFLD by influencing these processes. Studies conducted on mice lacking ERα, a specific type of estrogen receptor, have demonstrated disruptions in lipid metabolism. These disruptions include reduced fatty acid oxidation and increased lipid synthesis, leading to higher lipid accumulation in the liver compared to that of normal mice [18,19,20]. Moreover, inhibiting ERα downstream has been shown to contribute to heightened visceral fat accumulation and decreased energy expenditure in female mice [18,21]. In contrast, the administration of estrogen (E2) has been shown to stimulate the breakdown of fatty acids in the liver by upregulating the production of a protein called carnitine palmitoyltransferase 1 (CPT-1), which facilitates the transport of fatty acids into the mitochondria for oxidation [22]. Female mice that have undergone ovariectomy (OVX) experience improved insulin sensitivity and increased lipid export from the liver through the production of VLDL when estrogen is replaced [22]. 

Meta-analyses have indicated that men diagnosed with NAFLD have lower levels of serum testosterone, which can contribute to an accumulation of visceral adipose tissue and insulin resistance, both of which are associated with the development of hepatic steatosis [23,24]. The importance of androgen/androgen receptor (AR) signaling in suppressing NAFLD has been further supported by studies conducted on mice with hepatic AR deficiency, which yielded similar findings [24,25]. It has been discovered that androgen/AR signaling reduces the expression of sterol regulatory element-binding protein (SREBP), thereby inhibiting fatty acid synthesis. Additionally, it modulates phosphoinositide-3 kinase activity, which enhances insulin sensitivity [25].

Glucocorticoids have been found to enhance the activation of lipogenic genes, including fatty acid synthase (FASN) and acetyl-coA carboxylase 1 (Acaca) [26]. As a result, de novo lipogenesis is stimulated, while the secretion of VLDL is suppressed, ultimately leading to the development of hepatic steatosis [26]. Corticosterone also plays a significant role in this process.

### The Role of Steroid Hormones in Hepatic Inflammation and Fibrosis

The build-up of detrimental lipids leads to the creation of reactive oxygen species (ROS), stress within the endoplasmic reticulum (ER), cell demise, the liberation of damage-associated molecular patterns (DAMPs), and the discharge of inflammatory agents and extracellular vesicles. These elements activate an inflammatory reaction in Kupffer cells and a fibrotic reaction in hepatic stellate cells (HSC) [27,28]. The inflammatory process involves the recruitment of macrophages and neutrophils to the affected tissue, followed by the production of proinflammatory chemokines and cytokines such as interleukin 1 (IL-1) and tumor necrosis factor alpha (TNF-α) [29,30]. Additionally, these cytokines, in conjunction with IL-6, initiate the acute-phase response in liver inflammation [31]. The resulting liver inflammation can progress to fibrosis, ultimately resulting in cirrhosis [32].

Hepatic fibrosis is characterized by the accumulation of extracellular matrix. The transformation of hepatic stellate cells (HSCs) into myofibroblasts is a crucial factor in driving fibrosis, as it stimulates both proliferation and fibrogenesis [33]. Profibrogenic cytokines, such as vascular endothelial growth factor (VEGF), platelet-derived growth factor (PDGF), and transforming growth factor-β (TGF-β), have a significant impact on the activation of HSCs and the abnormal wound repair response in the liver [34,35,36].

## 2. Materials and Methods

To analyze the studies published on this topic, we conducted a literature review based on a scoping review design [37] on the effects of menopausal status, insulin resistance, and BMI on the prevalence of NAFLD. A scoping review has different characteristics compared to a systematic review, as it does not evaluate the literature to answer a specific question with a specific type of intervention but rather systematically analyzes the findings, identifies differences, and determines points for future research. The methodology does not differ significantly from a systematic review, although publications are not excluded based on the type of specific intervention. Throughout the scoping review we conducted, we made sure to adhere to the PRISMA-ScR guidelines. We followed these standardized guidelines with the aim of guaranteeing transparency, rigor, and reproducibility in both our methodology and report writing. The checklist from PRISMA-ScR helped us greatly; it assisted us in framing our research questions all the way through to data extraction and synthesis processes. We have taken a systematic approach by documenting our search strategy plus inclusion criteria and how data were extracted—this is intended to give readers a full view of what the literature looks like on this topic. Our adoption of PRISMA-ScR guidelines speaks volumes about our commitment to methodological rigor; moreover, it boosts credibility and makes our findings more useful for researchers, practitioners, and policymakers who rely on such information.

The aim of this review was to analyze the effects of menopausal status, insulin resistance, and BMI on the prevalence of NAFLD. A comprehensive search of published academic articles, media sources, and gray literature reports was performed to answer the question: ‘What is the effect of menopause, insulin resistance, and BMI on the prevalence of non-alcoholic fatty liver disease?’ The review was performed in three steps: (1) identifying the question and relevant literature; (2) selecting the literature; (3) charting, collating, and summarizing the information [37].

An extensive literature search of electronic databases was conducted (PubMed, Scopus, CINAHL, and Medline). The medical subject headings and entry terms ‘Menopause’, ‘Non-alcoholic fatty liver disease’, ‘Insulin resistance’, and ‘BMI’ were combined for the search. The search yielded 180 studies, of which 19 eventually met the requirements for inclusion (Figure 1). All publications in the English language published before the end of December 2023 were searched. An adjuvant search was performed on Google Scholar to ensure no relevant paper had been missed. Studies in the form of comments, author articles, case studies, review papers, book chapters, and those that did not report original data were not included.

To conduct a scoping review, it is necessary to include two distinct groups of participants: premenopausal women and postmenopausal women. This will allow for an evaluation of how menopausal status impacts the prevalence of non-alcoholic fatty liver disease (NAFLD) between the two groups. The diagnostic criteria for NAFLD should encompass abdominal ultrasound, magnetic resonance spectroscopy, and the fatty liver index. Notably, the fatty liver index has demonstrated satisfactory results compared to imaging techniques like abdominal ultrasound, magnetic resonance spectroscopy, and computed tomography, as reported by Castellana et al. [38]. Lastly, each study that is included in the review must provide outcomes specific to NAFLD patients in the premenopausal and postmenopausal groups, with a focus on insulin resistance and BMI, respectively.

The most common exclusion criteria stated in the studies reviewed were a medical record-established diagnosis of advanced disease (e.g., end-stage hepatic disease or cancer) or the presence of alcoholism, cognitive impairment, or a major psychiatric illness precluding active participation, such as a current schizophrenic/psychotic disorder, an eating disorder, bipolar disorder, addictive disorder, or personality disorder, being bedridden, or being under guardianship.

## 3. Results

Our search strategy identified 180 studies. After duplicates were eliminated, 132 papers required further full-text screening, of which 19 met the inclusion criteria and were analyzed in detail (Figure 1). Most articles were excluded because they did not measure the main outcome of the review. Two researchers independently summarized the results extracted from our literature review under the following headings: (1) Sample; (2) Duration; (3) Outcome. The most important data from the studies are summarized in Table A1, and the details of the results relevant to the outcomes of the review are discussed in the following sections under the headings outlined above.

### 3.1. Sample and Outcomes Regarding Menopausal Status

The included studies have different sample sizes ranging from 188 to 13,559 participants. All the studies were published between 2002 and 2023, with four of them performed in Korea, three in China, two in Brazil, two in the USA, one in Australia, one in Mexico, one in Ghana, and one in Italy.

Most of the studies concluded that there is an elevated prevalence of NAFLD in postmenopausal women. In a cross-sectional study conducted in Brazil, 188 postmenopausal women (aged ≥45 years and with amenorrhea ≥12 months) were included. Of the 188 women, 73 (38.8%) had NAFLD. In a cross-sectional study conducted in Ghana, the prevalence of NAFLD among postmenopausal women was 49.48%, which was higher than the 29.55% observed among premenopausal women. NAFLD showed a high prevalence among postmenopausal women, and the authors concluded that the presence of metabolic syndrome, abdominal obesity, and insulin resistance were indicators of risk for the development of NAFLD [39].

In a cross-sectional study by Wang et al. [40] the prevalence of NAFLD increased from 5.3% to 18.8% in women younger than 45 years versus women aged 45 to 55 years and rose to 27.8% in women older than 55 years. Accordingly, Gutierrez-Grobe et al. [41] found that the prevalence of NAFLD in premenopausal, postmenopausal, and PCOS patients was 32.2%, 57.9%, and 62%, respectively.

In the study by Ryu et al. [42] a higher prevalence of NAFLD was observed across menopausal stages (*p* < 0.05). After adjusting for age, center, BMI, smoking status, alcohol intake, physical activity, educational level, parity, and age at menarche, the odds ratios (95% CIs) for NAFLD comparing early transition, late transition, and post menopause to pre-menopause were 1.07 (0.68–1.67), 1.87 (1.23–2.85), and 1.67 (1.01–2.78), respectively. Correspondingly, Florentino et al. [43] observed that the prevalence of NAFLD was 37.1% (93/251) in postmenopausal women, 26.4% (14/53) in the group with hormone replacement therapy, and 39.9% (79/198) in the group without hormone replacement therapy.

Hamaguchi et al. [44] detected the incidence of NAFLD was 3.5% (28/802) in premenopausal women, 7.5% (4/53) in menopausal women, 6.1% (24/392) in postmenopausal women, and 5.3% (11/206) in women receiving hormone replacement therapy, while Park et al. [45] and Volzke et al. [46] concluded that menopausal status is associated with hepatic steatosis.

Among 13,559 women who participated in the National Health and Nutrition Examination Survey [47], the unadjusted NAFLD prevalence increased from 18.54% (1988–1994) to 21.36% (1999–2006) to 24.86% (2007–2014). The age-standardized prevalence of NAFLD in this cohort increased from 20.96% (1988–1994) to 26.19% (2007–2014).

Accordingly, Chen et al. [48] observed that the adjusted odds ratios (ORs) with 95% confidence intervals (CIs) for NAFLD among participants in the menopause transition period and postmenopausal period were 1.10 (0.89–1.37) and 1.28 (1.04–1.58), respectively, compared with women in the menstrual period, while Chung et al. [49] found the prevalence of NAFLD to be higher in postmenopausal women than in premenopausal women (27.2% versus 14.4%, *p* < 0.001).

Bao et al. [50] observed that regardless of whether the subjects were non-obese or obese, the prevalence of NAFLD was lower in non-menopausal subjects than in postmenopausal ones (non-obese: 20.74% vs. 45.26%, respectively, *p* < 0.0001), while Sanghavi et al. [51] stated that postmenopausal status was associated with an increased prevalence of hepatic steatosis (34% vs. 24%, *p* < 0.001).

Yang et al. [52] conducted a study with the objective of examining how gender and menopause are related to the extent of liver fibrosis. A total of 541 adult patients diagnosed with NASH were included in the analysis. The composition of the population consisted of 35.1% men, 28.4% premenopausal women, and 36.5% postmenopausal women. After adjusting for various confounding factors such as race, body mass index, diabetes/prediabetes, and hypertension, the adjusted cumulative odds ratio (ACOR) and 95% confidence interval (CI) for greater fibrosis severity were calculated. For postmenopausal women, the ACOR was 1.4 [0.9, 2.1] (*p* = 0.17), while for men it was 1.6 [1.0, 2.5] (*p* = 0.03), with pre-menopausal women used as the reference group. There was a borderline interaction between gender and age group, divided by the average age of menopause in the US (age 50), with a *p*-value of 0.08. Specifically, for patients under 50 years old, the ACOR and 95% CI for greater fibrosis severity in men compared to women was 1.8 [1.1, 2.9] (*p* = 0.02), while for patients 50 years and older, it was 1.2 [0.7, 2.1] (*p* = 0.59).

Yoneda et al. [53] conducted a study that specifically focused on women with non-alcoholic fatty liver disease (NAFLD) and a BMI below 30. Their objective was to compare the extent of hepatic fibrosis after menopause within this group. The study utilized data from the Japan Study Group of NAFLD (JSG-NAFLD) database, including a total of 419 women with biopsy-confirmed NAFLD. Among these participants, there were 90 premenopausal women and 329 postmenopausal women. The results revealed that, even among non-obese NAFLD patients, postmenopausal women exhibited more severe fibrosis compared to their premenopausal counterparts. This finding remained significant (*p* = 0.0266; odds ratio [OR]: 2.173) even after adjusting for factors such as hepatic inflammation, ballooning hepatocytes, BMI, impaired glucose tolerance/diabetes, and hypertension.

A meta-analysis of 12 studies [54] revealed a significant association between menopause and NAFLD, with a pooled OR of 2.37 (95% CI, 1.99–2.82; I^2^ = 73%). The association remained significant in a sensitivity meta-analysis of six studies that reported the association with adjustment for age and metabolic factors, with a pooled OR of 2.19 (95% CI, 1.73–2.78; I^2^ = 74%).

Two studies did not provide any evidence regarding the association of menopause with NAFLD risk. In a cross-sectional study by Park et al. [55], compared with normal menopausal women, early or late menopausal women had no significant differences in the odds ratios (ORs) of NAFLD: OR = 1.05, 95% CI, 0.83–1.32, and OR = 1.02, 95% CI, 0.75–1.39, respectively. Accordingly, in a study by Veronese et al. [56] the number of years since the onset of menopause was not associated with the severity of liver steatosis in NAFLD (*p* for trend = 0.74; Spearman correlation = 0.04; 95% CI: -0.09 to 0.17), whereas all the indexes of adiposity and the number of metabolic syndrome factors were associated with a higher liver steatosis score.

The occurrence of NAFLD is more prevalent among women who have experienced natural or surgically induced menopause compared to those who are still premenopausal [6,57]. The risk of NAFLD is similar in premenopausal women and those in the early stages of menopausal transition, but it increases by 87% during the late stages of menopausal transition, which are defined as periods of amenorrhea lasting 60 days or more [6].

Premature and particularly surgically induced menopause may have a more significant negative impact on liver health than natural menopause [57]. Women with NAFLD and premature menopause are nearly twice as likely to develop severe hepatic fibrosis compared to women with early or normal menopause, even after considering factors such as age, body measurements, hormone therapy use, and other metabolic diseases (odds ratio [OR] 1.9; 95% confidence interval [CI] 1.3–2.7) [58]. It appears that postmenopausal hypoestrogenism is a significant risk factor for the development of NAFLD. However, it is still unclear whether the decline in estrogen levels directly or indirectly contributes to the development of liver disturbances.

The impact of weight and metabolism on the liver may be more significant during the postmenopausal phase compared to the premenopausal phase, as indicated by previous research [59].

Earlier studies have shown that the rapid development of hypoestrogenism leads to a significant increase in the accumulation of liver fat and fibrosis in both ovariectomized animal models and in women who have undergone surgical menopause. However, the natural transition into menopause is a gradual process that involves the depletion of ovarian reserve and brings about changes in various hormonal and metabolic parameters. The association between menopause and non-alcoholic fatty liver disease (NAFLD) may be influenced by individual variations in estrogen, androgen, sex hormone-binding globulin (SHBG), and gonadotropin secretion [59].

The contradictory findings of various studies that have examined the hormonal patterns in postmenopausal women with simple steatosis and NASH have been a subject of investigation. It has been observed that postmenopausal patients with NASH have lower levels of estradiol compared to women with uncomplicated steatosis, as confirmed by histological analysis [60]. In humans, postmenopausal estrogen depletion is linked to a decrease in the number of CD4^+^ T cells and B cells, as well as an increase in the production of IL-1, IL-6, and TNF-α, accompanied by heightened sensitivity of immune cells to these cytokines. Furthermore, postmenopausal women with NAFLD experience suppressed cytotoxic activity in natural killer cells, as well as diminished phagocytic and antigen-presenting functions in macrophages and dendritic cells [61]. These findings suggest that estrogens also possess anti-inflammatory effects in humans, consistent with previous observations in animal models.

In a comprehensive study involving postmenopausal women in the United States, it was discovered that those with the highest estradiol levels had a greater likelihood of developing non-alcoholic fatty liver disease (NAFLD) compared to those with the lowest levels (OR 2.42, 95% CI 1.37–4.29). This association may be attributed to the higher prevalence of obesity and insulin resistance within the group under investigation [60]. Furthermore, increased levels of total testosterone contribute to the progression of NAFLD.

NAFLD is prevalent in women prior to menopause but decreases following menopause due to declining testosterone levels. Interestingly, postmenopausal women with NAFLD exhibit higher levels of bioavailable testosterone compared to healthy postmenopausal women. Conditions characterized by high androgen levels, such as PCOS, are linked to an elevated risk of NAFLD in women of reproductive age. However, diagnosing PCOS in older women poses challenges as it relies on historical information [52].

The development of NAFLD after menopause may be influenced by decreased levels of sex hormone-binding globulin (SHBG) [62,63,64]. A comprehensive analysis of existing research indicates that higher SHBG levels are linked to a reduced risk of NAFLD in women (OR 0.77, 95% CI 0.67–0.89) [65]. Given that SHBG plays a role in regulating hepatic lipogenesis by suppressing key lipogenic enzymes [65], it is plausible that it is involved in the pathogenesis of NAFLD after menopause. While postmenopausal hypoestrogenism exacerbates the development of NAFLD, the precise contributions of estrogens, androgens, SHBG, and their regulators to the onset of the disease remain uncertain.

### 3.2. Insulin Resistance, BMI and Other Factors

Bruno et al. [66] conducted a study that revealed notable disparities between individuals with non-alcoholic fatty liver disease (NAFLD) and those without it (the control group). The NAFLD group exhibited significantly elevated levels of blood pressure, waist circumference, body mass index, LDL cholesterol, triglycerides, and glucose (*p* < 0.05). Additionally, only the NAFLD group displayed insulin resistance, as indicated by the HOMA-IR values (6.1 ± 4.6 vs. 2.4 ± 1.4 in the control group, *p* < 0.05), according to the same study. Another investigation conducted by Wang et al. [40] reported that 48.4% of obese women were found to have NAFLD.

In the research conducted by Gutierrez-Grobe et al. [41], it was discovered that patients with non-alcoholic fatty liver disease (NAFLD) exhibited elevated levels of age, BMI, hip to waist ratio, fasting glucose, HOMA-IR, and insulin. Similarly, Florentino et al. [43] found that women diagnosed with NAFLD who did not use hormone replacement therapy demonstrated a higher prevalence of insulin resistance (HOMA-IR ≥3) (*p* < 0.05). Veronese et al. [56] observed a correlation between increased liver steatosis scores and various measures of adiposity, as well as the number of metabolic syndrome factors.

The results of a retrospective cohort study conducted in Korea, which included 728 premenopausal and 695 postmenopausal women, revealed distinct metabolic factors associated with non-alcoholic fatty liver disease (NAFLD) in each group. Among premenopausal women, low levels of HDL-cholesterol, central obesity, and insulin resistance, as measured by the homeostasis model assessment, were found to be significantly linked to an increased risk of NAFLD, according to the multivariate analysis. Conversely, postmenopausal women demonstrated a significant association between NAFLD risk and the presence of diabetes, elevated triglyceride levels, and central obesity [39].

## 4. Discussion

The objective of this scoping review was to examine how menopausal status, insulin resistance, and BMI impact the occurrence of NAFLD. Most studies indicated a notable increase in the prevalence of NAFLD among women who had reached menopause. However, two of the studies included in this review did not find a clear association between menopause and NAFLD.

The results align with a recent analysis of 12 cross-sectional studies, which conducted a systematic review and meta-analysis. This analysis revealed a 2.4-fold increase in the likelihood of non-alcoholic fatty liver disease (NAFLD) among individuals of menopausal status. Furthermore, even after adjusting for age and metabolic factors, this association remained significant in a sensitivity meta-analysis of six studies. The pooled odds ratio (OR) was calculated to be 2.19 (95% CI, 1.73–2.78; I^2^ = 74%) [54].

Non-alcoholic fatty liver disease is characterized by the buildup of fat in the liver, even in the absence of excessive alcohol consumption, certain genetic disorders, or the use of specific medications [28]. The presence of hepatic steatosis occurs when there is an imbalance between the accumulation and disposal of lipids in the liver. This can happen due to an increase in the uptake of fatty acids by the liver and a decrease in the disposal of lipids. The severity of hepatic steatosis is determined by the amount of fat present in the liver cells, which is graded from 0 (healthy, less than 5% fat) to 3 (severe, over 66% fat). Additionally, the percentage of hepatocytes containing visible triglycerides can be used to estimate the extent of hepatic steatosis [67].

Following menopause, there is a notable decrease in the levels of 17β-estradiol (E2), a hormone produced by the ovaries and present in the bloodstream. Extensive research has been conducted on the impact of estrogen deficiency on lipid metabolism in the liver using animal models. The correlation between menopause and the accumulation of fat in the liver implies that low levels of estradiol in the blood may be linked to non-alcoholic fatty liver disease (NAFLD).

The liver and adipose tissue rely heavily on estradiol, the most prevalent female reproductive hormone, to regulate lipid and glucose metabolism. In premenopausal women, the ovaries are the primary source of estradiol secretion [67]. However, postmenopause, ovarian estrogen secretion ceases, causing circulating estradiol levels to decrease to an average of around 10 pmol/L [68]. It is worth noting that although the ovaries no longer produce estradiol, small amounts are still synthesized by non-ovarian tissues [69,70]. The decline in circulating estradiol levels during natural menopause is linked to an increased susceptibility to NAFLD, type 2 diabetes, central obesity, and hypertriglyceridemia [71].

Postmenopausal women not only experience a lack of estrogen, but also an imbalance of androgens and a decline in sex hormone-binding proteins. These changes in hormone levels seem to coincide with an increase in abdominal fat mass, which is commonly observed in postmenopausal women. Additionally, age plays a significant role in the severity and progression of NAFLD, further exacerbating the situation.

The onset of post menopause is characterized by a decline in estradiol levels, which can have a detrimental effect on the expression of genes involved in lipid metabolism. This can ultimately lead to the development of obesity and metabolic disorders, which are key factors in the occurrence of NAFLD [64]. Postmenopausal women also experience an elevation in total serum cholesterol, triglycerides, and LDL levels, while the levels of high-density lipoprotein (HDL) cholesterol decrease [62]. Furthermore, there is a notable increase of 30–39% in small dense LDL levels, accompanied by an increase in small HDL subfraction 3 and a decrease in HDL subfraction 2 [64]. However, the association between menopause and triglyceride concentration remains unclear, suggesting that the rise in triglyceride levels may be attributed to the aging process [64]. In addition, obesity and dyslipidemia contribute to insulin resistance and the development of metabolic syndrome. Insulin resistance is linked to a decrease in adipose tissue lipolysis and an increase in the influx of free fatty acids into the liver, which predisposes individuals to NAFLD [66].

The systematic review by Venetsanaki & Polyzos [57], aimed to summarize studies linking menopause to NAFLD with a particular focus on potential therapeutic strategies. Preclinical and clinical studies seemed to favor the association between estrogen deficiency and NAFLD. Estrogen deficiency appeared to interact with relative androgen excess, increased visceral fat, and insulin resistance, and aging, changes that favor the presence and progression of NAFLD. Another issue highlighted by the review was the early management of excess androgens, as testosterone levels have been shown to increase rates of NAFLD in premenopausal women, and early management of excess testosterone may be useful in reducing the risk of NAFLD in women later in their reproductive lives.

During menopause, the liver’s ability to oxidize fatty acids decreases, and lipogenesis increases, leading to excess hepatic fat accumulation and, ultimately, inflammation [58]. Estrogen deficiency causes a redistribution of body fat, leading to visceral fat accumulation, which may influence the development and progression of NAFLD [6]. An experimental study in ovariectomized rats showed that a lack of estrogen stimulates fat accumulation in the liver, and that ovariectomized rats undergoing an endurance program had less fat in the liver and abdomen [61]. Lower ALT levels were also observed in patients with type 2 diabetes taking hormone replacement therapy (HRT) for 6 months [52].

In a cross-sectional study [72] of 541 individuals with biopsy-confirmed NASH, advanced fibrosis was more common in postmenopausal women (27.6%) than in men (22.2%) and premenopausal women (14.4%). Even after adjusting for covariates (admission site, race, degree of portal inflammation), women older than 50 years were more likely to have advanced fibrosis (OR 1.8, 95% CI 1.2–2.7). Lean postmenopausal women with NAFLD remained at increased risk of severe fibrosis compared with lean premenopausal women with NAFLD (OR 2.17, 95% CI 1.1–4.5) [59]. This suggests that menopause is associated with severe fibrosis and is partly independent of age or body fat composition.

Many studies have examined the use of HRT and the presence of NAFLD in postmenopausal women, leading to a better understanding of the effects of estrogen. A double-blind, randomized, placebo-controlled trial was conducted in 58 postmenopausal women with type 2 diabetes to determine whether HRT (1 mg estradiol/0.5 norethindrone [an androgen progesterone]) could improve liver enzyme levels. Among the 45 subjects who completed the study, subjects taking HRT for ≥6 months had lower levels of alanine aminotransferase (ALT), aspartate aminotransferase (serum concentrations of ASP), gamma-glutamyl transferase (GGT), and alkaline phosphatase were significantly reduced in those receiving placebo [52]. Among postmenopausal women with NAFLD, women who received HRT for ≥6 months (N = 14) had significantly lower waist circumference, GGT, ferritin, and insulin resistance scores significantly decreased compared with those of women who did not receive HRT (N = 79) [62].

A comprehensive study involving 251 postmenopausal women found that the occurrence of NAFLD was less prevalent among those who were using hormone replacement therapy (HRT) at a rate of 26.4%, compared to those who were not using HRT at a rate of 39.9%. HRT in this study encompassed the oral or transdermal administration of estrogen, a combination of estrogen and progesterone, and tibolone for a period of six months in postmenopausal women with an intact uterus or who had undergone a hysterectomy. While the incidence of NAFLD was lower in postmenopausal women using HRT, there was no significant association between the use of HRT and variables related to HRT (such as type, duration, and route of administration) in relation to NAFLD. NAFLD patients who were not taking HRT exhibited higher levels of GGT, ALT, ferritin, and insulin resistance. Additionally, women who were not taking HRT (regardless of NAFLD status) had a higher likelihood of being overweight, obese, and insulin resistant compared to women who were taking HRT [61]. Overall, these results suggest that HRT may partially protect against the development of postmenopausal NAFLD and attenuate the metabolic abnormalities associated with NAFLD.

## 5. Limitations

A wide number of studies were reviewed by the research that was carried out, many of which had a cross-sectional design. This type of study is limited in its ability to establish causal relationships between menopausal status, insulin resistance, BMI and NAFLD. As such, it can only provide a snapshot in time and does not indicate whether hormonal or metabolic changes precede or follow NAFLD.

The review delved into multiple studies differing in design, population, location, and diagnostic methods for NAFLD—a multitude of factors leading to inconsistency in results and hampering generalizability to postmenopausal women globally.

The meaning and prevalence of diseases such as NAFLD, insulin resistance, and menopausal condition can vary in different studies because different diagnostic criteria and methods are used to evaluate these conditions. Consequently, the comparability of results between studies may be hampered by disparities. An interesting gap in research lies in the lack of longitudinal data that observe how NAFLD advances while a woman undergoes menopause. To gain a detailed picture of the temporal relationships with NAFLD development and factors involved, it is essential for researchers to carry out comprehensive long-term studies.

Although certain studies have considered factors like age, lifestyle, and underlying conditions, it is important to recognize that there may be other variables that have not been measured that could affect outcomes. Dietary patterns, physical activity levels, genetic predisposition, and even socioeconomic status are players in NAFLD progression—but unfortunately, not all studies took these into account consistently.

The generalizability of findings can be compromised by a lack of representation of specific ethnic or socioeconomic groups, which limits the diversity within the study participants; this significantly impacts how applicable the results will be. The occurrence and impact of NAFLD differ greatly among populations, so more inclusive research should be conducted on these varying factors: their different manifestations vary from one population to another due to differing underlying causes.

The results of the review are based on the literature that was surveyed and included. Publication bias is a possibility when only studies showing significant findings are published, contributing to unintentional omissions of related studies. It is crucial to ensure an accurate appreciation of the review findings and guide successful efforts in future research by recognizing these limitations while concentrating on closing the gaps in the current literature. This can be achieved by focusing more effort on understanding why certain components have been left out and not blindly including them in future research works: in this way, we will be able to better appreciate the interplay between menopause metabolic changes with NAFLD, leading us towards more effective strategies for the prevention and treatment for NAFLD that will address all the metabolic complications that arise during the menopause as well.

## 6. Conclusions

In summary, this scoping review has revealed the intricate relationship between menopausal status, insulin resistance, and BMI that leads to NAFLD. Women exhibit a significantly higher prevalence of NAFLD post menopause, primarily due to hormonal changes—notably hypoestrogenism at this stage—which hamper the metabolic pathways that contribute to NAFLD. Moreover, heightened insulin resistance coupled with changes in body composition during menopause acts as a dual risk factor, signifying the complexity and multi-dimensional nature of NAFLD specifically among this female demographic post menopause.

While the review highlights these links, it underscores the varied nature of the findings as well as the need for more specialized probes. In future, priority should be given to longitudinal studies that would help establish causality and assist with exploring the intricacies surrounding these phenomena; they can consider personalized therapeutic interventions targeting different metabolic and hormonal profiles among postmenopausal women which may curtail wide propagation and progression of NAFLD in this population.

## Figures and Tables

**Figure 1 healthcare-12-01081-f001:**
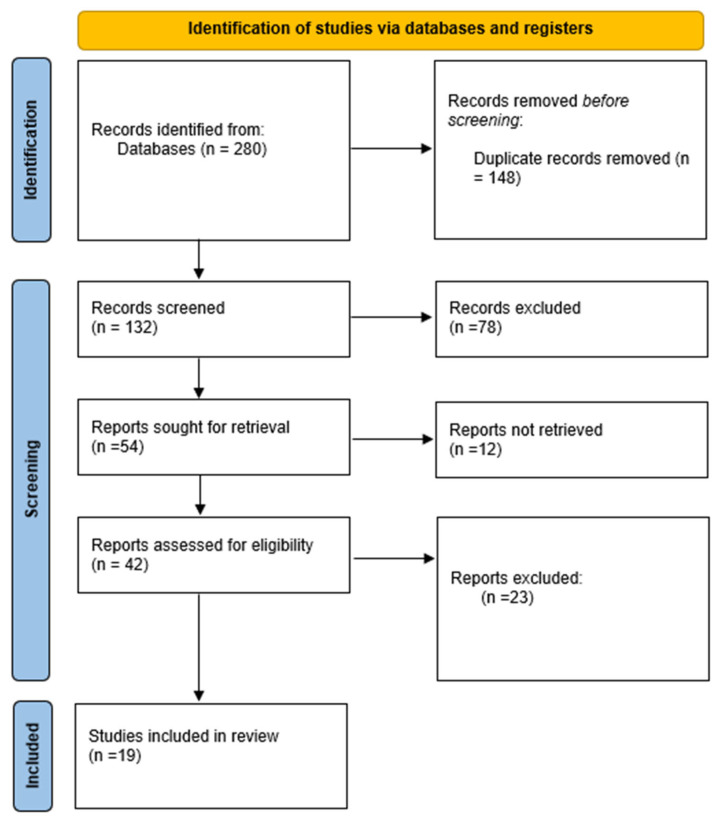
Study flow diagram.

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
