# Peer review of "The Effect of Menopausal Status, Insulin Resistance and Body Mass Index on the Prevalence of Non-Alcoholic Fatty Liver Disease"

_healthcare, 2024, doi:10.3390/healthcare12111081_

Round 1

Reviewer 1 Report

Comments and Suggestions for Authors

This manuscript addresses a timely topic and makes a relevant contribution to the field which elaborates a scoping review investigated the effects of menopausal status, insulin resistance and BMI on the prevalence of NAFLD. They also suggest that when treating women with NAFLD, their risk profile, hormonal status, age, and metabolic factors should be considered.

However, some major revisions are needed before it can be published.

1. The manuscript tries to achieve only few things for establishing elevated prevalence of NAFLD in postmenopausal women and also claimed insulin resistance and body mass index were significantly higher in women with NAFLD compared with women without NAFLD. I encourage the authors to provide more in-depth evidence on focused points (molecular mechanisms for effects of reproductive hormones on NAFLD).

2. The author only describes several studies related to the effect of menopausal status, insulin resistance and BMI on the prevalence of NAFLD but fails to justify any with proper molecular mechanism. Thereby it lacks the specificity as well as novelty. It could have been better if the author can provide the more specific crosstalk mentioning the mechanistic effects of reproductive hormones on NAFLD, possible therapeutic interventions with appropriate references.

 3. The author should include more studies with more patients for demographic and clinical characteristics to justify their claim regarding their proposed scoping review.

4. Inadequate and obsolete literature survey. It discusses findings in relation to some of the work in the field but ignores other important work. There are several repetitive and overlapping contents in the abstract, introduction and conclusion section which could have been avoided. Ensure that all references are current and relevant. It might also be beneficial to include more recent studies that have explored to contextualize your findings within the broader research landscape.

5. The conclusions cannot be justified on the basis of the rest of the paper. The author must rewrite the conclusion to justify their proposed scoping review.

6. The author must mention a separate paragraph mentioning the limitations of their study.

7. The language throughout the manuscript requires significant improvement by native English speakers.

8. The similarity content is very high for the manuscript which should be significantly reduced.

Comments on the Quality of English Language

1. The language throughout the manuscript requires significant improvement by native English speakers.

 2. The similarity content is very high for the manuscript which should be significantly reduced.

Author Response

Comments 1:  The manuscript tries to achieve only few things for establishing elevated prevalence of NAFLD in postmenopausal women and also claimed insulin resistance and body mass index were significantly higher in women with NAFLD compared with women without NAFLD. I encourage the authors to provide more in-depth evidence on focused points (molecular mechanisms for effects of reproductive hormones on NAFLD)

Response 1:

Thank you for pointing this out. We agree with this comment. Therefore, we have included the molecular mechanisms for the effects of reproductive hormones on NAFLD. These changes in the revised manuscript can be found – pages 2-3, lines 77-120

Comments 2:    The author only describes several studies related to the effect of menopausal status, insulin resistance and BMI on the prevalence of NAFLD but fails to justify any with proper molecular mechanism. Thereby it lacks the specificity as well as novelty. It could have been better if the author can provide the more specific crosstalk mentioning the mechanistic effects of reproductive hormones on NAFLD, possible therapeutic interventions with appropriate references.

Response 2: Agree. We have, accordingly, modified the text to emphasize this point. These changes in the revised manuscript can be found – pages 2-3, lines 124-139

Comments 3:    The author should include more studies with more patients for demographic and clinical characteristics to justify their claim regarding their proposed scoping review.

Response 3: Agree. We have, accordingly, modified the text to emphasize this point. These changes in the revised manuscript can be found – pages 6-7, lines 248-270.

Comments 4:    Inadequate and obsolete literature survey. It discusses findings in relation to some of the work in the field but ignores other important work. There are several repetitive and overlapping contents in the abstract, introduction and conclusion section which could have been avoided. Ensure that all references are current and relevant. It might also be beneficial to include more recent studies that have explored to contextualize your findings within the broader research landscape.

Response 4: Agree. We have, accordingly, modified the text to emphasize this point. These changes in the revised manuscript can be found –lines 257-279.

Comments 5:    The conclusions cannot be justified on the basis of the rest of the paper. The author must rewrite the conclusion to justify their proposed scoping review.

Response 5: Agree. The conclusion has been rewritten in order to justify the proposed scoping review. We have, accordingly, modified the text to emphasize this point. These changes in the revised manuscript can be found – page 12, lines 523-536.

Comments 6:    The author must mention a separate paragraph mentioning the limitations of their study.

Response 6: Agree. A separate paragraph mentioning the limitations of the study has been written. These changes in the revised manuscript can be found – pages 11- 12, lines 484-521.

Comments 7:    The language throughout the manuscript requires significant improvement by native English speakers.

Response 7: Agree. The language throughout the manuscript has been improved by a native English speaker.

Comments 8:    The similarity content is very high for the manuscript which should be significantly reduced.

Response 8: Agree. The similarity of the content has been significantly reduced.

Reviewer 2 Report

Comments and Suggestions for Authors

I thank the manuscript's authors, "The effect of menopausal status, insulin resistance and BMI on the prevalence of NAFLD," for their time writing this manuscript. The aim of this manuscript was to analyze the effects of menopausal status, insulin resistance and BMI on the prevalence of NAFLD. The authors conducted a scopus review.

I ask the authors to revise their manuscript's text and correct the following.

The authors must include bibliographical references after the last word, followed by the point. For example, Line 28: adults. [1] = adults [1].

Line 42: 40    and = 40 and = In the pdf there seems to be a double or triple space.

Line 142: (p <0.05).     After = (p <0.05). After = In the pdf there seems to be a double or triple space.

Line 156: (2007-20140. [28] = (2007-20140) [28].

Line 168: P<0.0001) = p<0.0001) = The "p" of the p-value must be written in lowercase.

Line 175: I2 or I2

Line 177: cross- sectional = cross-sectional

Line 194:  (P<0.05) =  (p<0.05) = The "p" of the p-value must be written in lowercase.

The authors do not mention that they respected PRISMA in their drafting of the scoping review. They should therefore revise the text and follow the rules of PRISMA.

Author Response

I thank the manuscript's authors, "The effect of menopausal status, insulin resistance and BMI on the prevalence of NAFLD," for their time writing this manuscript. The aim of this manuscript was to analyze the effects of menopausal status, insulin resistance and BMI on the prevalence of NAFLD. The authors conducted a scopus review.

I ask the authors to revise their manuscript's text and correct the following.

The authors must include bibliographical references after the last word, followed by the point. For example, Line 28: adults. [1] = adults [1].

            We agree with the comment. Done.

Line 42: 40    and = 40 and = In the pdf there seems to be a double or triple space.

            Done.

Line 142: (p <0.05).     After = (p <0.05). After = In the pdf there seems to be a double or triple space.

Done.

Line 156: (2007-20140. [28] = (2007-20140) [28].

            Done.

Line 168: P<0.0001) = p<0.0001) = The "p" of the p-value must be written in lowercase.

Done.

Line 175: I2 or I2

              Done.

Line 177: cross- sectional = cross-sectional

Done.

Line 194: (P<0.05) = (p<0.05) = The "p" of the p-value must be written in lowercase.

          Done.

The authors do not mention that they respected PRISMA in their drafting of the scoping review. They should therefore revise the text and follow the rules of PRISMA.

          Done.

Reviewer 3 Report

Comments and Suggestions for Authors

This scoping review attempts to answer the question of how do insulin resistance, menopause, and BMI affect the prevalence of NAFLD. The authors comprehensively review the related body of literature which suggests that these is in fact a relation between insulin resistance, menopause, and BMI on one hand and increasing NAFLD prevalence.

1. A multisociety Delphi consensus statement has renamed NAFLD to metabolic-dysfunction associated liver disease (MASLD) halfway through 2023. I would recommend updating article to reflect those changes or add "recently called..", with the caveat that the literature up to that point uses NAFLD so it would make sense to included that as well in the search strategy. There is also a Met-ALD combination category including alcohol related liver disease. All of these are under the bigger umbrella of steatotic liver disease.

2. (line 100) Would suggest not using "must", and would also mention biopsy even though that is being used less frequently nowadays. NAFLD may be diagnosed by blood tests that show liver inflammation markers and may be used to calculate special scores, imaging tests such as US, CT, MRI, US-elastography, and biopsy which is a less frequently used and much more invasive.

3. Would be beneficial to highlight what this review adds to the literature in terms of the relationship between menopausal status and NAFLD/MASLD since there's already a meta analysis looking into this.

4. Were sex hormones assessed in relation to NAFLD/MASLD prevalence or was it only in context of menopause? If not, what was the rationale?

Comments on the Quality of English Language

Minor edits concerning quality of English language are required to make the sentences flow smoother.

Author Response

  1. A multisociety Delphi consensus statement has renamed NAFLD to metabolic-dysfunction associated liver disease (MASLD) halfway through 2023. I would recommend updating article to reflect those changes or add "recently called..", with the caveat that the literature up to that point uses NAFLD so it would make sense to included that as well in the search strategy. There is also a Met-ALD combination category including alcohol related liver disease. All of these are under the bigger umbrella of steatotic liver disease.

We decided not to add this term at this point.

2. (line 100) Would suggest not using "must", and would also mention biopsy even though that is being used less frequently nowadays. NAFLD may be diagnosed by blood tests that show liver inflammation markers and may be used to calculate special scores, imaging tests such as US, CT, MRI, US-elastography, and biopsy which is a less frequently used and much more invasive.

We changed it.

3. Would be beneficial to highlight what this review adds to the literature in terms of the relationship between menopausal status and NAFLD/MASLD since there's already a meta analysis looking into this. Added.

4. Were sex hormones assessed in relation to NAFLD/MASLD prevalence or was it only in context of menopause? If not, what was the rationale? The sex hormones assessed in relation to NAFLD and were added to the text.

Round 2

Reviewer 1 Report

Comments and Suggestions for Authors

Most of the concerns have addressed and it sounds better now.

Comments on the Quality of English Language

still there are scope for improvement!